# Effect of the Granularity of Cubic Boron Nitride Vitrified Grinding Wheels on the Planar Technical Blades Sharpening Process

**DOI:** 10.3390/ma15227989

**Published:** 2022-11-11

**Authors:** Bartosz Zieliński, Krzysztof Nadolny, Wojciech Zawadka, Tomasz Chaciński, Wojciech Stachurski, Gilmar Ferreira Batalha

**Affiliations:** 1Espersen Koszalin Sp. o.o., Mieszka I 29, 75-124 Koszalin, Poland; 2Department of Production Engineering, Faculty of Mechanical Engineering, Koszalin University of Technology, Racławicka 15-17, 75-620 Koszalin, Poland; 3Institute of Machine Tools and Production Engineering, Lodz University of Technology, Stefanowskiego 1/15, 90-537 Lodz, Poland; 4Escola Politecnica da Universidade de Sao Paulo, Sao Paulo 01246-904, Brazil

**Keywords:** grinding, sharpening, cBN abrasive grain, vitrified bond, cutting blade, food processing

## Abstract

The most widely used method for shaping technical blades is grinding with abrasive tools made of cubic boron nitride (cBN) grains and vitrified bond. The goal of this work was to determine the effect of grinding wheel grain size (cBN grain number according to FEPA standards: B126, B181 and B251), kinematics (grinding with the circumference, face and conical surface of the wheel) and feed rate (*v_f_* = 100; 150; 200 mm/min) on the effects of the grinding process evaluated by the cutting force of the blade after machining *F*, blade surface texture parameters (*Sa*, *St*, *Smvr*, *Str*, *Sdq*, *Sdr* and *Sbi*) as well as blade surface morphology. An analysis of output quantities showed that grinding wheels made of B181 cBN grains are most favorable for shaping planar technical blades of X39Cr13 steel in the grinding process.

## 1. Introduction

The characteristic of production operations carried out in the modern food industry, especially in its areas related to fish processing, as described in detail in the work of Sen [1], Boziaris [2] and Borda and co-authors [3], requires the use of several (sometimes complex) processing operations resulting in the efficient separation of raw fish material. One such operation is the processing of the raw material to remove unwanted elements (fins, heads, bones and backbone, among others) and to give it the proper shape and dimensions. Part of the processing is skinning, described in the work of Hall [4,5], which involves separating the fillet (a flap of meat without bones or skin) from the fish. In this case, technological machines operating in an automatic or manual cycle are usually used. They are equipped with single technical blades or their assemblies containing, in some cases, as many as 6–8 blades or more, and they may be stationary with respect to the fed raw material or may perform reciprocating motion.

The cutting blades used in the food processing industry are mainly made of carbon and alloy tool steels, high-speed steels as well as stainless steels, as characterized in the work of Colás and Totten [6]. The latter are most used in the food industry and are approved for contact with food. Since the skinning process is one of the key operations that determine the target weight of an intermediate product as well as its shape and dimensions (as indicated by the work of Hall [4]), food processors place great emphasis on maintaining a short process time and high efficiency. This process can be disrupted by the occurrence of several unfavorable factors, making it difficult or (in some cases) impossible to conduct. Most often, they are related to the condition of the cutting tool surface. This condition is significantly influenced by, among others, the following:factors conducive to strong corrosive interactions (including a humid work environment, the use of alkaline compounds and phosphates to decontaminate and protect the raw material from microorganisms and the use of nitric and phosphoric acid to remove post-production sludge);factors that promote mechanical wear, mainly the change in the geometry of the cutting edges of blades because of the machining process (e.g., dulling of cutting edges);factors that promote a change in the properties of the material, leading to an increase in its susceptibility to deformation.

The most widely used method for shaping technical blades is abrasive machining [7], including the process of grinding with abrasive tools made of cBN grains and vitrified bond, as characterized in the work of Jackson and Davim [8].

The superhard abrasive cBN was discovered in 1957 as a by-product of diamond synthesis research. The first commercial product appeared in 1968, when General Electric Co. (Boston, MA, USA) introduced cBN under the name Borazon. Today, cBN is produced by many companies around the world, and its share in the group of all abrasive tools is steadily increasing [9].

The cBN grains are characterized by sharp vertices and cutting edges and have a more developed surface than diamond. Cubic boron nitride is used in the shaping and finishing grinding processes of steel, cast iron, stainless and alloy steels; non-ferrous metals; and for machining hard-to-cut high-speed steels with high carbide content. Unlike diamond, it shows resistance to the chemical effects of iron, cobalt and nickel at high temperatures. All these advantages make it used, despite its high price, for machining those groups of materials that cannot be satisfactorily ground with conventional abrasives (due to their hardness) or diamond (due to its limited resistance to temperature and chemical wear) [10,11]. Table 1 shows the chemical composition and properties of regular boron nitride [8,9,10,11,12].

The development of grinding technology with cBN abrasive tools made it possible to significantly increase the achievable process efficiency (more than 10-fold higher grinding ratio *G* values compared to conventional grinding [9,12]) by using significantly higher cutting speeds. On the other hand, by increasing the feed rate, it was possible to significantly reduce (compared to conventional grinding) the manufacturing times when machining hard-to-cut materials (reducing the cycle time by 80%), which include stainless steels.

Recently, there has been a noticeable trend toward the use of cBN abrasive tools with a vitrified bond [13,14,15]. This type of bond chemically binds the cBN grains, obtaining an open porous structure of the abrasive layer, thus making it possible to take full advantage of the benefits of cBN by easily renewing the cutting ability of grinding wheels within the dressing procedure [9,11]. The vitrified bond also allows for low workpiece heating and low grinding forces. The increased interest in this type of grinding wheels was also conditioned by the development of a new generation of grinding machines and peripheral equipment (sensor-controlled dressing units, optimized coolant supply).

There is a lack of research results in the directional literature on the use of vitrified grinding wheels with cBN grains in the grinding process of technical blades. It can be assumed that this knowledge is the know-how of tool manufacturers who do not make it public. This limits the development of abrasive machining techniques applied to shape technical blades used in the food processing industry. The goal of this work was to expand the knowledge in this area to include the issue of selecting the abrasive grain size of cBN tools used for grinding planar technical blades.

## 2. Materials and Methods

The goal of this experimental study was to determine the effect of:grinding wheel grain size (cBN grain number according to Federation of European Producers of Abrasives standards: B126, B181 and B251),process kinematics (grinding with the circumference of the wheel; grinding with the face of the wheel; grinding with the conical surface of the wheel),and feed rate (*v_f_* = 100; 150; 200 mm/min), on the results of the grinding evaluated by the cutting force of the blade after machining *F*, blade surface texture parameters as well as blade surface morphology indicated by microscopic observation.

The following set of parameters was selected for surface texture evaluation (software: TalyMap Silver 4.1.2, Digital Surf, Besançon, France):arithmetic mean deviation of the surface *Sa*;total height of the surface *St*;mean void volume ratio *Smvr*;density of summits of the surface *Sds*;texture aspect ratio of the surface *Str*;root-mean-square slope of the surface *Sdq*;developed interfacial area ratio *Sdr*;bearing index *Sbi*.

In the experimental research, a special stand equipped with a 5-axis computerized numerical control (CNC) grinder was used (Figure 1a). This grinder (described in detail in work [16]) allowed for a wide range of grinding parameter adjustments, variable process kinematics as well as rigid workpiece clamping (Figure 1b). In this case, the workpieces were planar technical knives with dimensions of 459.5 mm × 12.3 mm × 0.6 mm (Kuno Wasser GmbH, Solingen, Germany, for Steen F.P.M. International, Kalmthout, Belgium) made of high-carbon martensitic stainless steel X39Cr13 (Figure 1c) with the axial outline shown in Figure 1d. The tests were carried out without repetition, but each time, the grinding wheel was dressed, thus restoring its cutting ability. After the grinding operation, it was necessary to remove the rewind from the edge of the blade. To obtain full repeatability of this procedure, a special fixture was developed and made, in which the rewind was removed using a leather set on a flat wooden support, which was moved several times along the machined edge of the blade in a rectilinear feed motion (Figure 1e).

For the study, type 5A1 grinding wheels from INTER-DIAMENT (Grodzisk Mazowiecki, Poland) made of cBN abrasive grains and vitrified bond with the technical parameters shown in Table 2 were selected.

The grinding wheels used were characterized by the same features of construction (type of bond, structure, hardness, type of abrasive grains) in addition to the dimensions of the cBN grains. Grinding wheels with cBN grains numbered (according to FEPA standards) B126, B181 and B251 were selected for the study. Table 3 shows a summary of the designations and characteristic dimensions of the abrasive micrograins used according to various industry standards.

The grinding process was resized in three variants of the kinematic system, in which the machining was carried out, respectively, with the circumference (Figure 2a), face (Figure 2b) and specially shaped conical surface of the grinding wheel (Figure 2c). 

The angular position of the grinding wheel relative to the workpiece changed in each of the listed kinematic variations of the grinding process. Figure 3 shows the designations of the three angles used to define the position of the grinding wheel during the tests.

After the tests, the knives’ cutting forces were determined after grinding on a special test stand shown in Figure 4 (descriptions of the test stand and cutting tests’ methodology are provided in the work [18]), the values of selected surface texture parameters of the blades were determined (on a position for contact measurements of surface texture equipped with a stylus profilometer Hommel-Tester T8000 by Hommelwerke GmbH, Villingen-Schwenningen, Germany [19]) and the morphology of the blades was analyzed (using a digital measuring microscope Dino-Lite Edge AM7515MT8A, AnMo Electronics Corporation, New Taipei City, Taiwan [20,21,22]).

The study used an experimental planning methodology, which resulted in a research plan and the number of repetitions of each point in the plan *n* = 3. Table 4 presents a detailed summary of the conditions and parameters of the experimental tests carried out.

## 3. Results and Discussion

To simplify and shorten the description of the tested grinding wheels, a set of abbreviated designations was adopted based on the most important difference between the wheels: the number of the cBN abrasive grains used in their construction. The analysis of the test results was divided into three sections relating to the three main groups of output factors determined during the experimental tests conducted:the cutting force *F* recorded during the test of cutting through the test specimen with the knife after sharpening on a special test stand (Section 3.1);the values of selected surface texture parameters: *Sa*, *St*, *Smvr*, *Str*, *Sdq*, *Sdr* and *Sbi* (Section 3.2);an analysis of the morphology of the blade surface after grinding assessed by microscopic observations (Section 3.3).

### 3.1. Analysis of Cutting Force Values F

For each combination of variable grinding conditions, three blades were shaped (number of repetitions *n* = 3), followed by one cutting force measurement for each of them on a special measuring station described in more detail in the work [18]. The next three figures (Figure 5, Figure 6 and Figure 7) show the same measurement results separately for each kinematic variation of the grinding process: grinding with the periphery of the grinding wheel (Figure 5), grinding with the face of the wheel (Figure 6) and grinding with the conical chamfer (Figure 7).

To simplify the analysis of the obtained results of cutting force measurements, a chart was also drawn up showing its average values determined from measurements of three blades shaped with the same grinding process parameters (Figure 8).

In addition, the average cutting force values determined from blade measurements for the same kinematic variation of the grinding process (Figure 9a) and for an equal value of the feed rate *v_f_* (Figure 9b) were also compiled. In the greatest generalization, the measured force values are presented in the form of average values calculated from the data from all tests by cBN abrasive grain size and shown in Figure 9c.

The measurement results obtained showed that the planar blade surfaces shaped with a B126 grinding wheel required the use of the relatively highest force values to cut the test specimens. Such a correspondence was observed both when the results of measurements for the same kinematic variation of the grinding process (Figure 9a) and for the same value of the feed rate *v_f_* (Figure 9b) were compiled. As a result, the summed value determined from all trials (18.62 N) was about 19% higher with respect to the summed values of the cutting force obtained for blades ground with B181 (15.67 N) and B251 (15.72 N) grinding wheels, as shown in the chart in Figure 9c. In the case of cutting force results measured during cutting tests with blades ground with B181 and B251 grinding wheels, the differences in values were much smaller and showed no significant trend. This was also evident in the total values, which were very similar (Figure 9c).

As a result, based on the obtained results of cutting force measurements, it can only be concluded that the least favorable results were obtained during grinding with the B126 grinding wheel. To find an explanation for the obtained results of cutting force measurements, further analyses were carried out on the surface texture (Section 3.2) and microscopic images (Section 3.3) of shaped technical blades.

### 3.2. Analysis of the Blade Surface Texture

Nine blades were selected for surface microtopography measurements, in such a way that a surface texture evaluation was conducted on one representative for each combination of variable grinding process parameters.

Figure 10 shows a collection of axonometric views of 1.0 × 1.0 mm surface microtopographies measured for nine selected planar blades. The figure also includes a list of the adopted designations of the measured blades and a description of the conditions under which their grinding process was carried out.

The analyzed surface texture parameters are then presented graphically (Figure 11, Figure 12, Figure 13, Figure 14, Figure 15, Figure 16, Figure 17 and Figure 18) in the form of charts showing the values determined for the surface of each blade and the average values calculated for the data grouped by abrasive grain size (B126, B181 and B251).

The measured microtopographs clearly showed regular machining marks oriented according to the direction of displacement of active abrasive grains on the grinding wheel and resulting from the combination of the rotational and feed motion of the tool (Figure 10). Local defects (single vertices) and scratches were also visible, especially on the surface of blades shaped with B126 (Blade no. 1–3) and B251 (Blade no. 7–9) grinding wheels (Figure 10).

The calculated values of the amplitude parameters *Sa* and *St* indicated very low roughness of the blade’s surface (*Sa* = 0.0516–0.2060 µm; *St* = 0.667–3.340 µm): Figure 11 and Figure 12. At the same time, significant (up to 4-fold) differences in the values of these parameters for individual surfaces were visible. However, it seems that this should be explained by the random nature of the surface shaped in a highly stochastic grinding process and the presence of few surface defects (summits or voids), which significantly affected the values of the amplitude parameters.

The aforementioned local and sparse defects were also clearly exposed in the values of the *Smvr* parameter, describing the mean void volume ratio (Figure 13). On the other hand, they no longer had such a significant effect on the density of summits of the surface *Sds*, represented in the number of vertices per mm^2^ of the evaluated surface (Figure 14). 

Between 4527 and 7563 vertices per mm^2^ were identified on the measured blades, with the lowest values attributed to blades ground with the B181 grinding wheel. The values of the second of the analyzed surface texture parameters belonging to the group of spatial parameters (texture aspect ratio of the surface *Str*) indicated favorable surface characteristics (lower values) of blades shaped with B181 and B251 grinding wheels (Figure 15).

Parameters belonging to the group of hybrid parameters (*Sdq*–Figure 16 and *Sdr*–Figure 17) did not reveal a significant trend in their variability, allowing us to assess the influence of the type of abrasive grain size on their values. The situation was different in the case of the value of the bearing index *Sbi* (Figure 18), which was included in the group of functional surface texture parameters. In this case, the beneficial effect on the bearing capacity of the surface of using the B181 grinding wheel during its shaping became apparent. 

The higher bearing capacity can be interpreted as a higher material proportion positively influencing the functional properties of surfaces working in contact with the counter surface, as was also the case in the cutting process.

Summarizing the obtained results of measurements of the surface texture of nine technical blades (Figure 10, Figure 11, Figure 12, Figure 13, Figure 14, Figure 15, Figure 16, Figure 17 and Figure 18), it can be concluded that the differences between the values of the surface texture parameters calculated for the surfaces shaped by the three compared types of grinding wheels were relatively small. However, considering the values of the bearing index *Sbi* (Figure 18), a grinding wheel with B181 grains was selected as the wheel that allowed the blade surface to be shaped with the most favorable functional characteristics.

### 3.3. Analysis of the Blade Surface Morphology

Figure 19, Figure 20 and Figure 21 show microscopic views of the blade surfaces of planar knives shaped by grinding wheels with cBN abrasive grains numbered B126 (Figure 19), B181 (Figure 20) and B251 (Figure 21) recorded with a Dino-Lite Edge AM7515MT8A digital measuring microscope (AnMo Electronics Corporation, New Taipei City, Taiwan). 

An analysis of these images confirmed the presence of characteristic features of the blades’ surfaces, found earlier on the basis of an analysis of their microtopography. They showed clear machining traces in the form of scratches, the direction of which corresponded to the kinematics of the abrasive grains shaping the surface in the different varieties of the grinding process (peripheral, face and grinding using the grinding wheel with the conical chamfer). In addition, a few surface defects in the form of cracks pointing in a direction different from the dominant one were observed on microscopic images. They could have been formed during the procedure of removing the rewind of the blade, in which the leather tool was moved along the blade, and it is possible that the removed chips of the workpiece material as well as fragments of abrasive grains or bond remaining on the machined surface accumulated on it.

Microscopic images also revealed the presence of a few irregular pits in the shaped surface. It seems that these defects occurred most frequently on the surface of blades ground with the B251 grinding wheel (Figure 21).

The size of the cBN micrograins determined the number of active abrasive grains present on the grinding wheel active surface (the surface involved in the removal of material). As the size of the grain increased, this number decreased, which, with unchanged machining parameters, translates into an increase in the cross-sectional area of the machined layer attributable to a single abrasive grain. The results obtained show that when the cBN grains of the smallest size (B126) were used, the shaped surface of the blade was characterized by features resulting in the relatively highest value of the cutting force *F* required to separate the material with their use. An analysis of the surface texture of the blade and its morphology allowed us to determine that this may have been caused by the way the blade was shaped resulting from grinding the two side surfaces of the blade. When very fine grains are used, the number of contacts of the active cutting vertices with the workpiece surface increases, with many of these grains having a negative rake angle. This results in an increase in the share of friction in machining (compared to grinding with grinding wheels with larger grain sizes). This leads to an increase in the heat flux generated during machining while hindering its dissipation through coolant, which can reach the grinding wheel–workpiece contact zone in relatively smaller intergranular spaces. The heat penetrates quite easily into the workpiece material due to its small thickness in the machining area, and this can result in an increase in the share of the plastic deformation phenomenon of the shaped edge. As a result, the shaped blade geometry imposes the highest force in the cutting process.

Increasing the size of abrasive grains allows the coolant to reach the grinding zone more easily, while reducing the number of abrasive grains directly involved in material removal. This, in turn, increases the cross-sectional area of the machined layer by a single grain and can result in an increase in the roughness of the machined surface as well as an increased intensity of grain wear phenomena (dulling and vertices chipping). The results obtained indicate grinding wheels with cBN grains of B181 as tools that allow shaping the surface of blades with the highest load capacity (comparing the value of the bearing index *Sbi* parameter). The study shows that it was this type of grain that made it possible to obtain a favorable compromise between the described factors determining the achieved performance properties of the ground blades.

## 4. Conclusions

The experimental studies conducted have made it possible to formulate several specific conclusions, which are listed below.

The results of the measurements showed that the blade surfaces shaped with grinding wheels made of cBN grains numbered B126 required the use of the relatively highest force values to cut the test specimens: the total value determined from all tests (18.62 N) was about 19% higher with respect to the total cutting force values obtained for blades ground with grinding wheels made of cBN grains numbered B181 (15.67 N) and B251 (15.72 N).The microtopographs of the blade surfaces clearly showed regular machining traces oriented according to the direction of movement of active abrasive grains on the grinding wheel surface and resulting from the combination of the rotational and feed motion of the tool.Surface texture microtopographs also revealed local defects (single vertices) and scratches, occurring mainly on the surface of blades shaped with an abrasive wheel with B126 (blade #1–3) and B251 (blade #7–9) grains.The differences between the values of surface texture parameters calculated for the surfaces shaped by the three types of grinding wheels compared (B121, B181 and B251) were relatively small. However, considering the values of the bearing index *Sbi*, a grinding wheel with B181 grains was selected as the wheel that allowed the shaping of a blade surface with the most favorable functional characteristics.An analysis of microscopic images of the blade surface of planar knives confirmed the characteristic features of the blades’ surfaces, previously found based on the analysis of their microtopography. Few surface defects in the form of cracks oriented in a direction different from the dominant one were observed on the microscopic images, which may be the result of the procedure of removing the rewind of the blade. Microscopic images also presented a few irregular pits in the shaped surface occurring most abundantly on the surface of blades ground with the B251 grinding wheel.Considering the overall results of the cutting force *F* value measurements, surface texture analysis and microscopic observations, it was found that grinding wheels made of cBN grains of B181 (of the granularity included in the study) were most favorable for shaping the planar technical blades of X39Cr13 steel in the grinding process.

## Figures and Tables

**Figure 1 materials-15-07989-f001:**
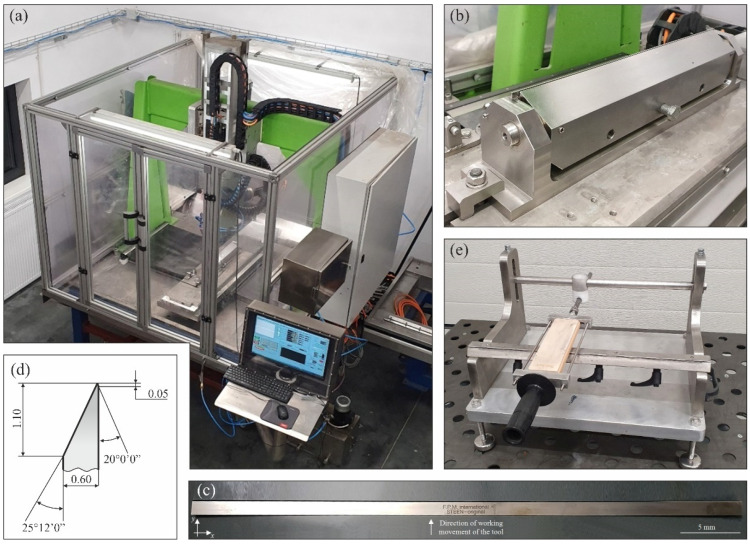
General view of the five-axis CNC special grinder for sharpening technical blades (**a**), view of workpiece clamping fixture (**b**), general view (**c**) and geometric dimensions of the axial outline (**d**) of the planar technical blade (workpiece) as well as view of a special device for removing rewind from blade after grinding (**e**).

**Figure 2 materials-15-07989-f002:**
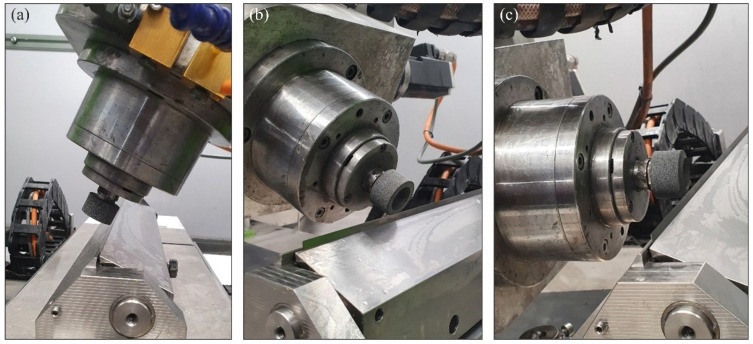
View of the working zone for grinding with the circumference (**a**), face (**b**) and conical surface of the grinding wheel (**c**).

**Figure 3 materials-15-07989-f003:**
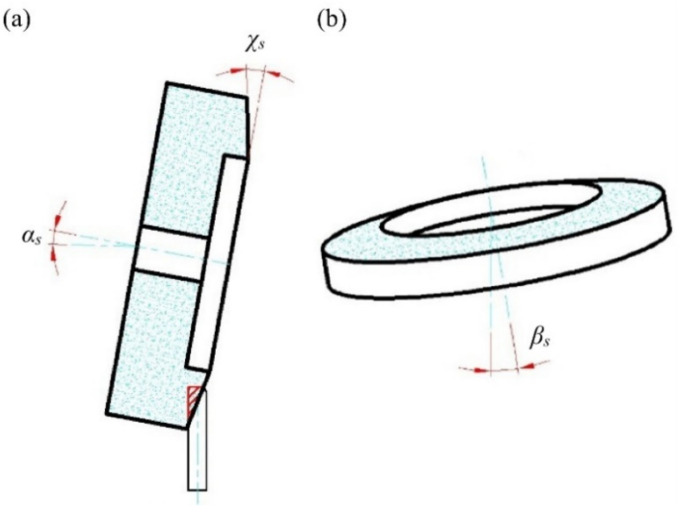
Diagram of the angular positioning of the grinding wheel in three planes with respect to the planar blade: (**a**) side view; (**b**) front view.

**Figure 4 materials-15-07989-f004:**
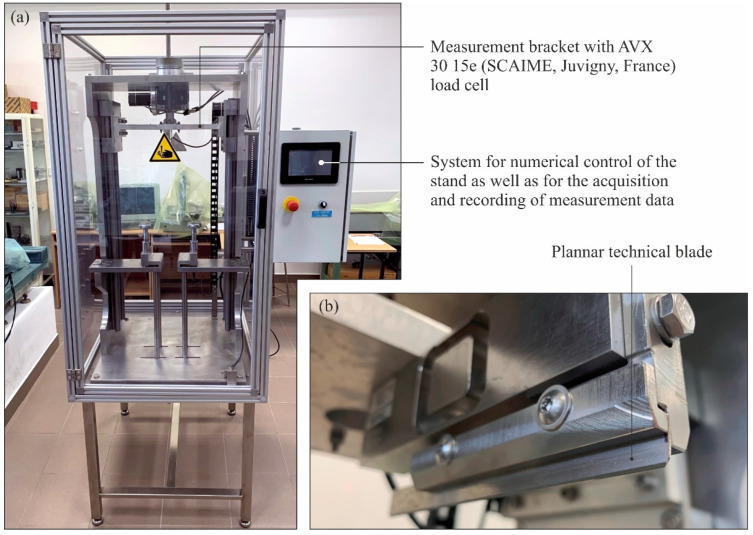
General view (**a**) and working space (**b**) of the special station for measuring the cutting force of planar technical blades.

**Figure 5 materials-15-07989-f005:**
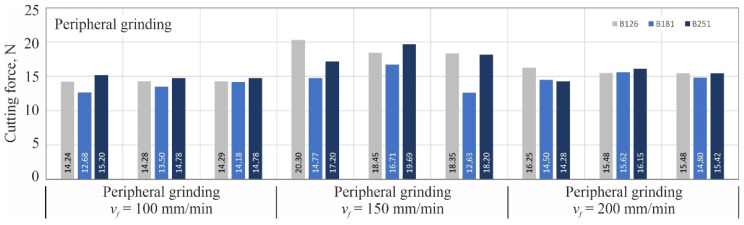
Results of cutting force measurements of blades shaped by peripheral grinding with cBN abrasive grains B126, B181 and B251 at feed rates *v_f_* of 100; 150; 200 mm/min.

**Figure 6 materials-15-07989-f006:**
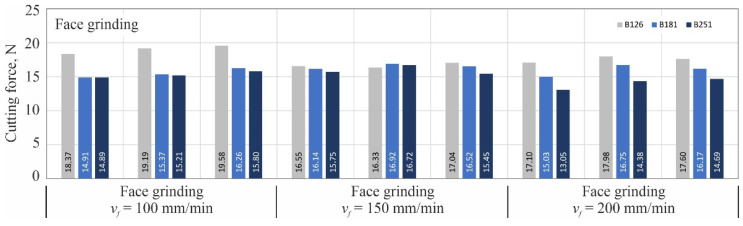
Results of cutting force measurements of blades shaped by face grinding with cBN abrasive grains B126, B181 and B251 at feed rates *v_f_* of 100; 150; 200 mm/min.

**Figure 7 materials-15-07989-f007:**
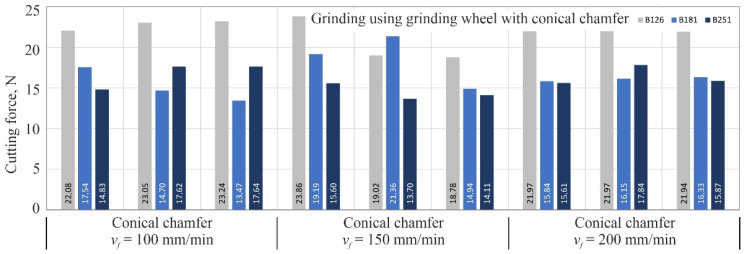
Results of cutting force measurements with blades shaped by grinding wheel with conical chamfer made of cBN abrasive grains B126, B181 and B251 at feed rates *v_f_* of 100; 150; 200 mm/min.

**Figure 8 materials-15-07989-f008:**
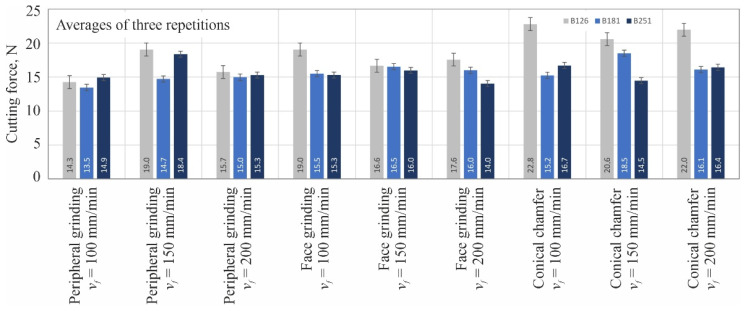
Average cutting force values determined from measurements of three blades shaped with the same grinding process parameters using grinding wheels with cBN abrasive grains B126, B181 and B251 (error bars represent the standard error equal to the standard deviation *σ* divided by the square root of the total number of samples).

**Figure 9 materials-15-07989-f009:**
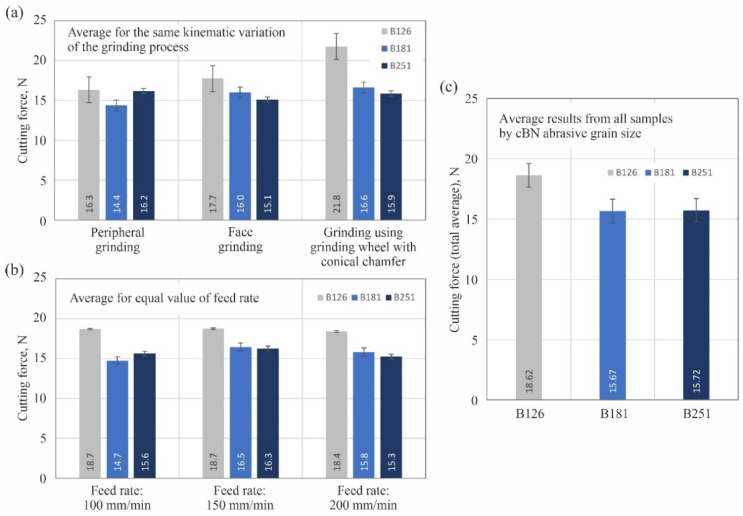
Average cutting force values determined from measurements of blades shaped using grinding wheels with cBN abrasive grains numbered B126, B181 and B251: (**a**) for the same kinematic variation of the grinding process; (**b**) for equal value of feed rate *v_f_*; (**c**) average results from all samples by cBN abrasive grain size (error bars represent the standard error equal to the standard deviation *σ* divided by the square root of the total number of samples).

**Figure 10 materials-15-07989-f010:**
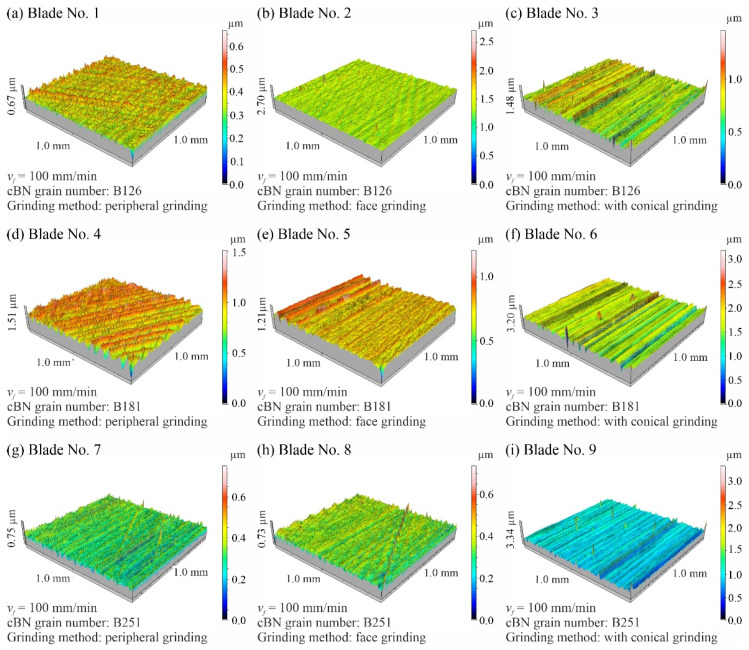
View of surfaces microtopographs (1.0 × 1.0 mm) recorded for nine selected planar blades on a measuring position equipped with a Hommel-Tester T8000 stylus profilometer from Hommelwerke GmbH (Villingen-Schwenningen, Germany): (**a**) blade no. 1; (**b**) blade no. 2; (**c**) blade no. 3; (**d**) blade no. 4; (**e**) blade no. 5; (**f**) blade no. 6; (**g**) blade no. 7; (**h**) blade no. 8; (**i**) blade no. 9.

**Figure 11 materials-15-07989-f011:**
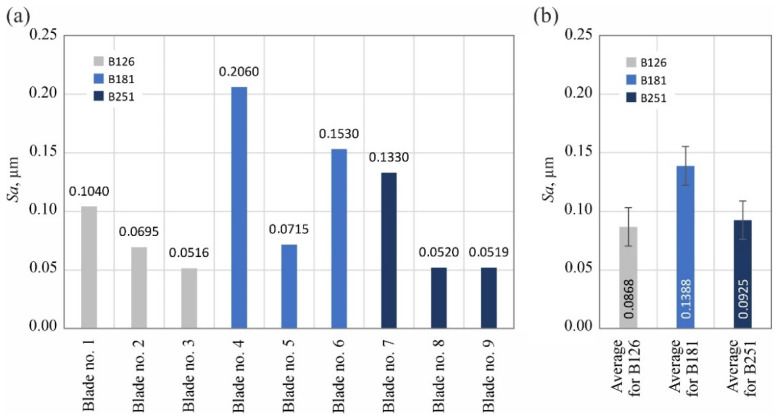
Changes in the arithmetic mean deviation of the surface *Sa*: (**a**) results for nine blades selected for surface texture measurements; (**b**) average values determined by the size of the cBN abrasive grain of the grinding wheel used (error bars represent the standard error equal to the standard deviation *σ* divided by the square root of the total number of samples).

**Figure 12 materials-15-07989-f012:**
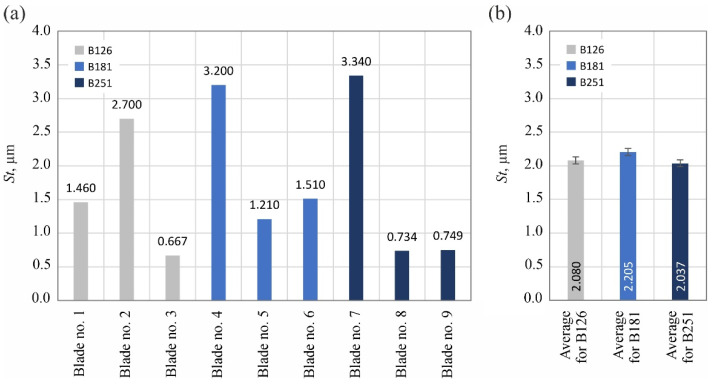
Changes in the total height of the surface *St*: (**a**) results for nine blades selected for surface texture measurements; (**b**) average values determined by the size of the cBN abrasive grain of the grinding wheel used (error bars represent the standard error equal to the standard deviation *σ* divided by the square root of the total number of samples).

**Figure 13 materials-15-07989-f013:**
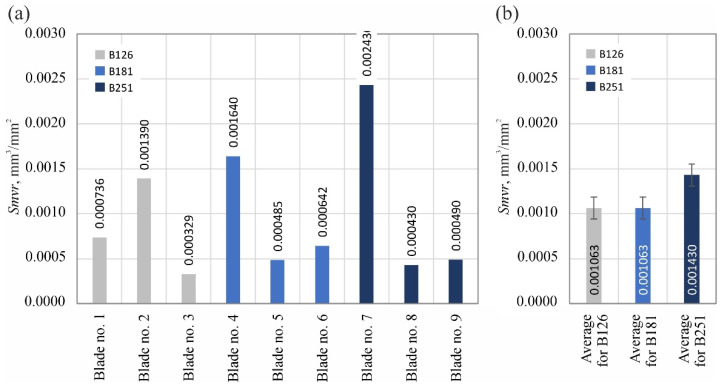
Changes in the mean void volume ratio *Smvr*: (**a**) results for nine blades selected for surface texture measurements; (**b**) average values determined by the size of the cBN abrasive grain of the grinding wheel used (error bars represent the standard error equal to the standard deviation *σ* divided by the square root of the total number of samples).

**Figure 14 materials-15-07989-f014:**
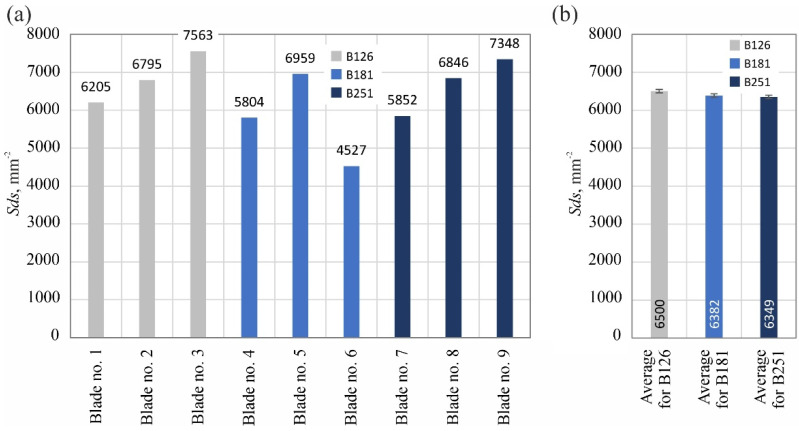
Changes in the density of summits of the surface *Sds*: (**a**) results for nine blades selected for surface texture measurements; (**b**) average values determined by the size of the cBN abrasive grain of the grinding wheel used (error bars represent the standard error equal to the standard deviation *σ* divided by the square root of the total number of samples).

**Figure 15 materials-15-07989-f015:**
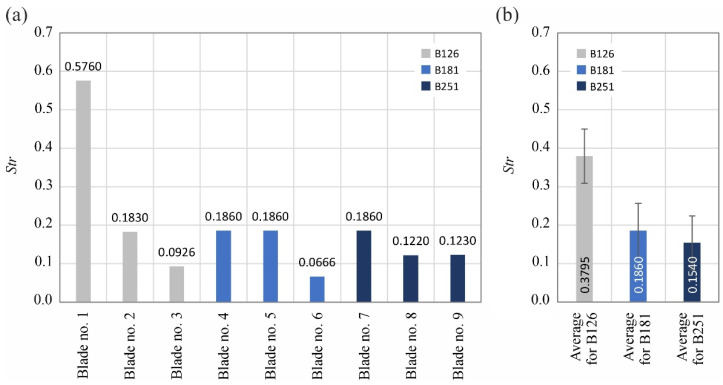
Changes in the texture aspect ratio of the surface *Str*: (**a**) results for nine blades selected for surface texture measurements; (**b**) average values determined by the size of the cBN abrasive grain of the grinding wheel used (error bars represent the standard error equal to the standard deviation *σ* divided by the square root of the total number of samples).

**Figure 16 materials-15-07989-f016:**
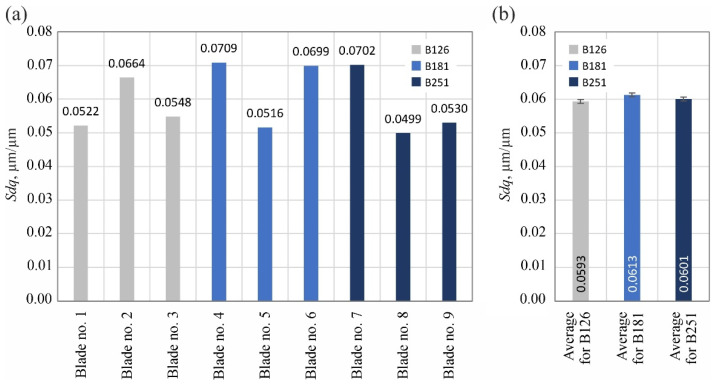
Changes in the root-mean-square slope of the surface *Sdq*: (**a**) results for nine blades selected for surface texture measurements; (**b**) average values determined by the size of the cBN abrasive grain of the grinding wheel used (error bars represent the standard error equal to the standard deviation *σ* divided by the square root of the total number of samples).

**Figure 17 materials-15-07989-f017:**
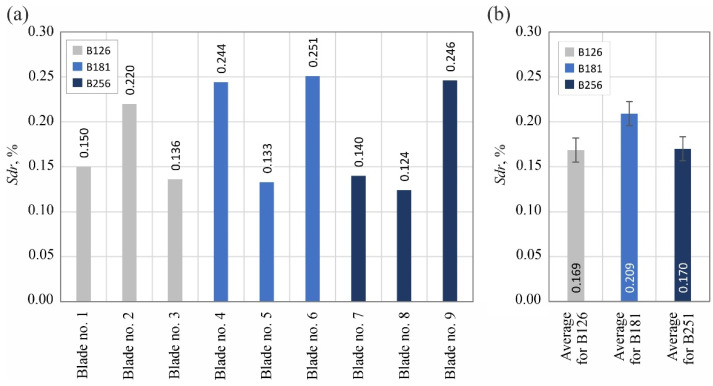
Changes in the developed interfacial area ratio *Sdr*: (**a**) results for nine blades selected for surface texture measurements; (**b**) average values determined by the size of the cBN abrasive grain of the grinding wheel used (error bars represent the standard error equal to the standard deviation *σ* divided by the square root of the total number of samples).

**Figure 18 materials-15-07989-f018:**
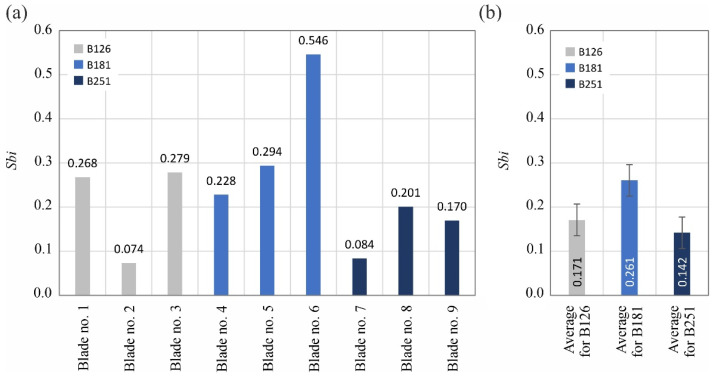
Changes in the bearing index *Sbi*: (**a**) results for nine blades selected for surface texture measurements; (**b**) average values determined by the size of the cBN abrasive grain of the grinding wheel used (error bars represent the standard error equal to the standard deviation *σ* divided by the square root of the total number of samples).

**Figure 19 materials-15-07989-f019:**
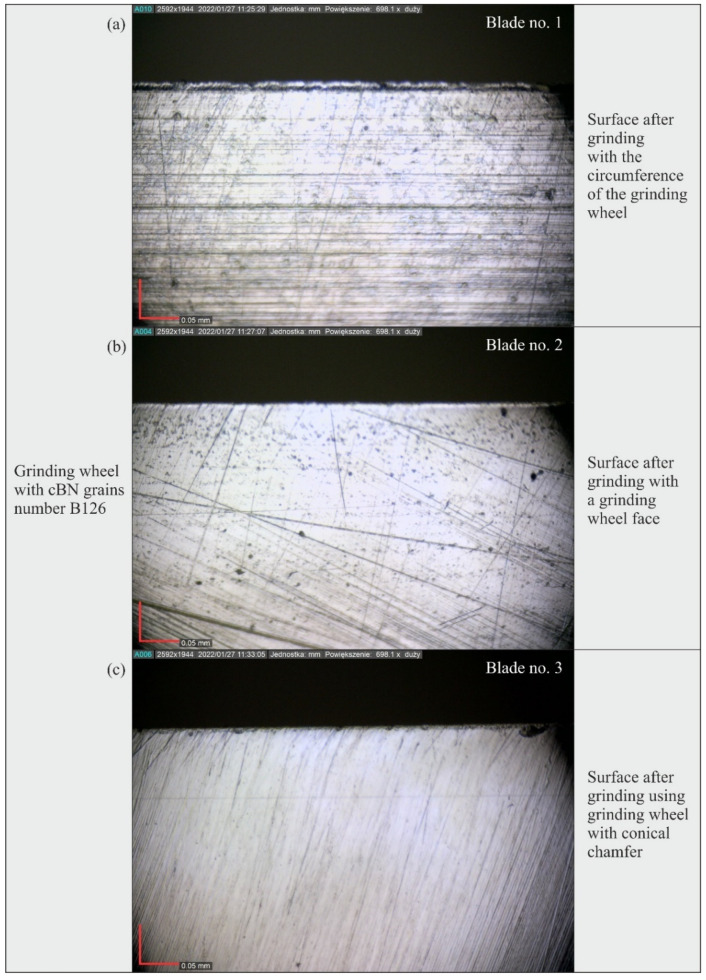
Microscopic views of the blade surface of planar knives shaped using grinding wheels with cBN abrasive grains numbered B126 at a magnification of approximately 700×: (**a**) surface after peripheral grinding (blade no. 1); (**b**) surface after face grinding (blade no. 2); (**c**) surface after grinding using grinding wheel with conical chamfer (blade no. 3).

**Figure 20 materials-15-07989-f020:**
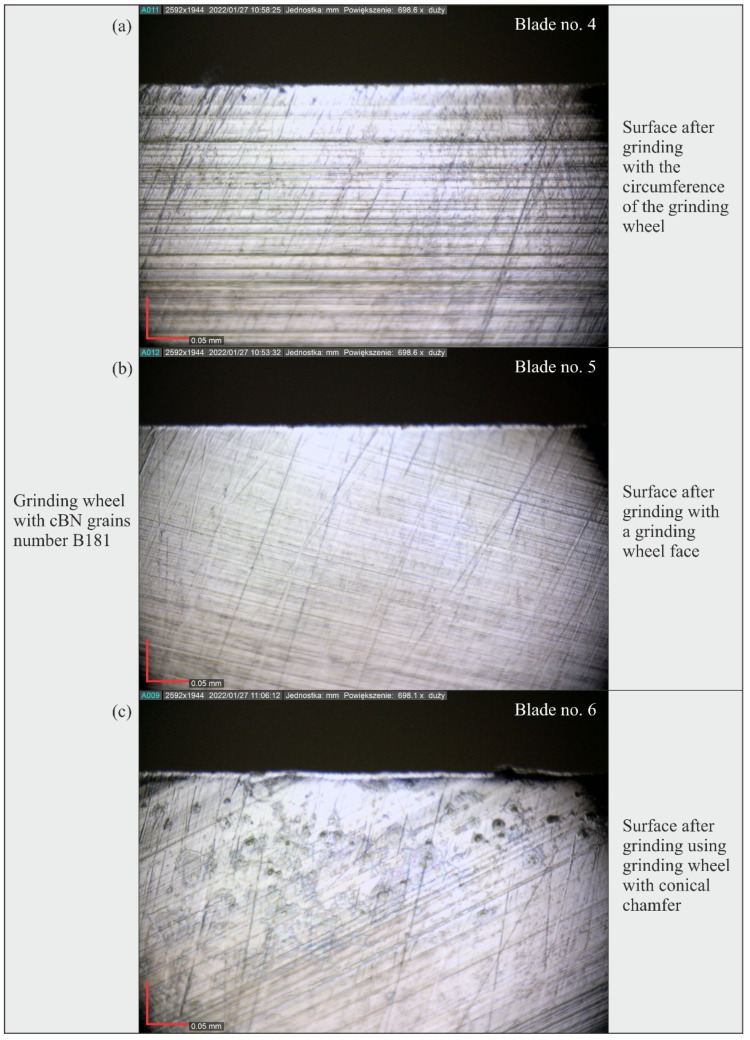
Microscopic views of the blade surface of planar knives shaped using grinding wheels with cBN abrasive grains numbered B181 at a magnification of approximately 700×: (**a**) surface after peripheral grinding (blade no. 4); (**b**) surface after face grinding (blade no. 5); (**c**) surface after grinding using grinding wheel with conical chamfer (blade no. 6).

**Figure 21 materials-15-07989-f021:**
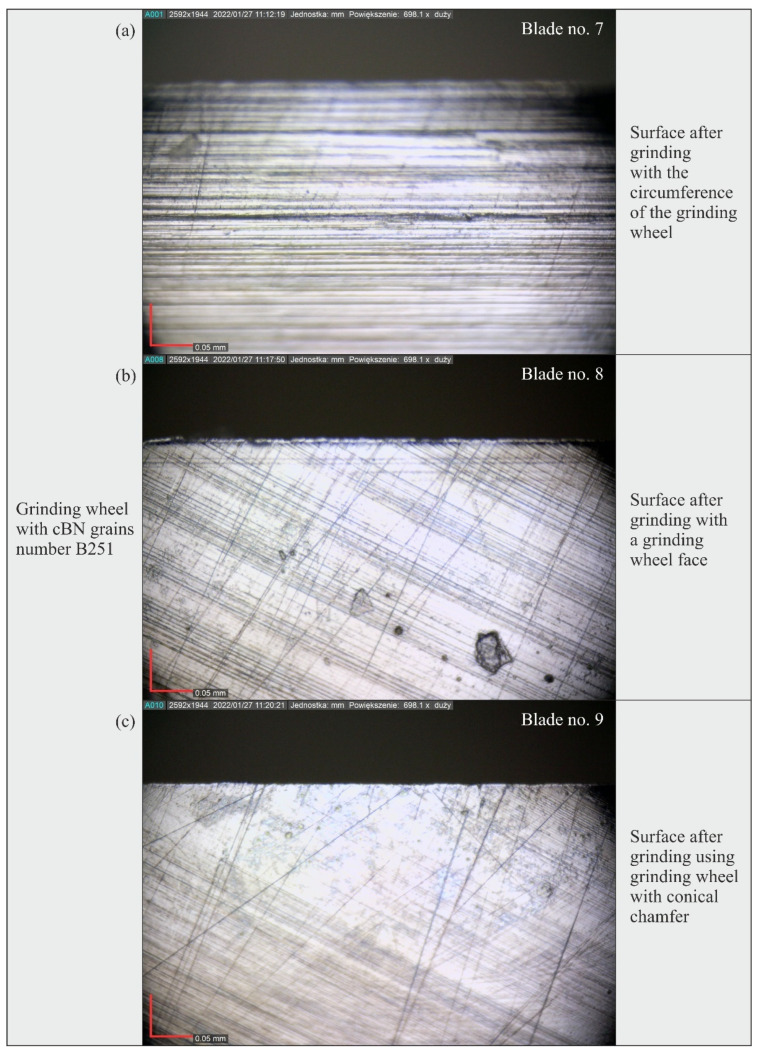
Microscopic views of the blade surface of planar knives shaped using grinding wheels with cBN abrasive grains numbered B251 at a magnification of approximately 700×: (**a**) surface after peripheral grinding (blade no. 7); (**b**) surface after face grinding (blade no. 8); (**c**) surface after grinding using grinding wheel with conical chamfer (blade no. 9).

**Table 1 materials-15-07989-t001:** Composition and properties of cBN abrasives [8,9,10,11,12].

Full Name	Cubic Boron Nitride (cBN)
**Chemical composition**	~43.6% B ~56.4% N
**Size of the crystal**	From ~10 μm (monocrystalline) to <1 μm (microcrystalline)
**Shape**	Blocky (hexagonal) or irregular, very sharp
**Specific density**	3.48 g/cm^3^
**Knoop HK hardness**	42–54 GPa
**Critical stress intensity factor *K_Ic_***	3.7 MPa·m^1/2^
**Coefficient of friction (hardened steel)**	0.19
**Thermal conductivity coefficient *λ***	240–1300 W/m·K

**Table 2 materials-15-07989-t002:** Designation and main technical parameters of the grinding wheels [17].

Designation	Dimensions					Composition			
	*D*, mm	*P*, mm	*T*, mm	*H*, mm	*F*, mm	Grain Size According to FEPA, µm	Grain Concentration, Carat/cm^3^	Degree of Hardness	Bond
5A1 35 × 25 × 10/22 × 15 B181 V240 SV	35	22	25	10	15	180/150	4.18	Medium	Vitrified
5A1 35 × 25 × 10/22 × 15 B181 V240 SV	35	22	25	10	15	180/150	4.18	Medium	Vitrified
5A1 35 × 25 × 10/22 × 15 B251 V240 SV	35	22	25	10	15	250/212	4.18	Medium	Vitrified
**Diagram of grinding wheel construction with designations of basic dimensions**	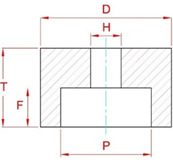								

**Table 3 materials-15-07989-t003:** Summary of designations and dimensions of characteristic abrasive micrograins according to various industry standards [17].

FEPA Grain Number	FEPA Grain Size	Grain Size US Standard ASTM E-11
B126	125/106 µm	120/140 mesh
B181	180/150 µm	80/100 mesh
B251	250/212 µm	60/70 mesh

**Table 4 materials-15-07989-t004:** Specification of parameters and conditions of experimental tests.

Process	Rectilinear Grinding of Flat Surfaces	
**Test stand**	Specialized five-axis CNC grinding machine for shaping knife blades with low rigidity	
**Workpiece**	Planar knives designed for the process of skinning flat fish made of high-carbon martensitic X39Cr13 stainless steel (Kuno Wasser GmbH, Solingen, Germany for Steen F.P.M. International, Kalmthout, Belgium)	
**Grinding wheel**	5A1 35 × 25 × 10/22 × 15 B126 V240 SV; 5A1 35 × 25 × 10/22 × 15 B181 V240 SV; 5A1 35 × 25 × 10/22 × 15 B251 V240 SV (INTER-DIAMENT, Grodzisk Mazowiecki, Poland)	
**Dressing parameters of the grinding wheel**	Dresser: M1039/D 1.00 ct (Dialeks, Pruszków, Poland) Rotational speed of grinding wheel in dressing: *n_sd_* = 32,000 rpm Feed rate during dressing: *v_fd_* = 0.00165 m/s Dressing allowance: *a_d_* = 0.03 mm Number of dressing passes: *i_d_* = 2	
**Grinding parameters**	**Constant input quantities**	**Variable input quantities**
	Rotational speed of grinding wheel: *n_s_* = 32,000 rpm Allowance for rough passage: *a_e rough_* = 0.10 mm Number of roughing passes for both phases of the blade: 1 Allowance for sparking-out passage: *a_e sparking-out_* = 0.02 mm Number of sparking-out passes for both phases of the blade: 1 Direction of rotation of the grinding wheel: right–in the direction of the axis of symmetry of the knife (to the blade) Angular positioning of the grinding wheel while grinding with the circumference of the wheel: *α_s_* = 25°, *β_s_* = 0°, *χ_s_* = 0° Angular positioning of the grinding wheel while grinding with the face of the wheel: *α_s_* = 65°, *β_s_* = 0°, *χ_s_* = 0° Angular positioning of the grinding wheel while grinding with the conical surface of the wheel: *α_s_* = 85°, *β_s_* = 5°, *χ_s_* = 20°	Grinding wheels: 5A1 35 × 25 × 10/22 × 15 B126 V240 SV (Abbreviated designation: B126) 5A1 35 × 25 × 10/22 × 15 B181 V240 SV (Abbreviated designation: B181) 5A1 35 × 25 × 10/22 × 15 B251 V240 SV (Abbreviated designation: B251) Grinding kinematics: grinding with the circumference of the wheel; grinding with the face of the wheel; grinding with the conical surface of the wheel Longitudinal feed velocity of the grinding wheel: *v_f_* = 100; 150; 200 mm/min
**Cooling conditions**	Flooding (WET) cooling using a low-pressure circular nozzle with expenditure *Q* = 1.75 dm^3^/min. Coolant: 5% water–oil emulsion of Cimtech^®^ M26 oil by CIMCOOL^®^ Fluid Technology forming part of Milacron LLC (Cincinnati, OH, USA)	

## Data Availability

Not applicable.

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
