# Peer review of "Effect of the Granularity of Cubic Boron Nitride Vitrified Grinding Wheels on the Planar Technical Blades Sharpening Process"

_materials, 2022, doi:10.3390/ma15227989_

Round 1

Reviewer 1 Report

In this manuscript, the effects of grinding wheel grain size, process kinematics, and feed rate on the cutting force of the blade were systematically evaluated, the surface parameters and surface morphology were also observed. Comments and suggestions are follows.

(1) The big concern of the manuscript is that, there is few further discussion or analysis after the observation of the 81 tests. Please add some valuable discussions at the end of Section 3, e.g. what we can learn from so many results? what kind of blade should be selected and why?

(2) In "surface texture parameters", the usage of "texture" in this manuscript may be not good. The presented several parameters is for profile description.

(3) The data in Figure 5 and Figures 6-8 is the same results, it is better to delete Figure 5 or add it into supplementary material.

(4) The data in Table 7 and Figures 12-15 is the same results, it is better to delete Figures 12-15 or add them into supplementary material.

(5) Table 5 is not necessary, the abbreviation B126, B181, and B251, can be added into Table 4.

(6) "cubic boron nitride cBN grains" should be "cubic boron nitride (cBN) grains" or "cBN grains"

(7) The first long sentence in Section 2 is not clear.

Author Response

The Authors wish to thank all the Reviewers for their time spent on prepare review of the manuscript “Effect of the granularity of cubic boron nitride vitrified grinding wheels on the planar technical blades sharpening process”. All the valuable comments, suggestions and hints were very helpful in improving the readability of the text and in improving of its scientific quality. Below the detailed responses upon these subsequent comments were given. The modified or added text in the manuscript was highlighted in red.

REVIEWER 1

In this manuscript, the effects of grinding wheel grain size, process kinematics, and feed rate on the cutting force of the blade were systematically evaluated, the surface parameters and surface morphology were also observed. Comments and suggestions are follows.

Reviewer comment 1:

The big concern of the manuscript is that, there is few further discussion or analysis after the observation of the 81 tests. Please add some valuable discussions at the end of Section 3, e.g. what we can learn from so many results? what kind of blade should be selected and why?

Authors response:

At the end of Section 3, a summary analysis of the results of the obtained studies was added in accordance with the reviewer's comment:

“The size of the cBN micrograins determines the number of active abrasive grains present of the grinding wheel active surface (the surface involved in the removal of material). As the size of the grain increases, this number decreases, which, with unchanged machining parameters, translates into an increase in the cross-sectional area of the machined layer attributable to a single abrasive grain. The results obtained show that when the cBN grains of the smallest size (B126) were used, the shaped surface of the blade was characterized by features resulting in the relatively highest value of the cutting force F required to separate the material with their use. Analysis of the surface texture of the blade and its morphology allows us to determine that this may have been caused by the way the blade was shaped resulting from grinding the two side surfaces of the blade. When very fine grains are used, the number of contacts of the active cutting vertices with the workpiece surface increases, with many of these grains having a negative rake angle. This results in an increase in the share of friction in machining (compared to grinding with grinding wheels with larger grain sizes). This leads to an increase in the heat flux generated during machining while hindering its dissipation through coolant, which can reach the grinding wheel-workpiece contact zone in relatively smaller intergranular spaces. The heat penetrates quite easily into the workpiece material due to its small thickness in the machining area, and this can result in an increase in the share of the plastic deformation phenomenon of the shaped edge. As a result, the shaped blade geometry imposes the highest force in the cutting process.

Increasing the size of abrasive grains allows coolant to reach the grinding zone more easily, while reducing the number of abrasive grains directly involved in material removal. This, in turn, increases the cross-sectional area of the machined layer by a single grain and can result in an increase in the roughness of the machined surface as well as an increased intensity of grain wear phenomena (dulling and vertices chipping). The results obtained indicate grinding wheels with cBN grains of B181 as tools that allow shaping the surface of blades with the highest load capacity (comparing the value of the bearing index Sbi parameter). The study shows that it was this type of grain that made it possible to obtain a favorable compromise between the described factors determining the achieved performance properties of the ground blades.”

Reviewer comment 2:

In "surface texture parameters", the usage of "texture" in this manuscript may be not good. The presented several parameters is for profile description.

Authors response:

Roughness parameters determined from a single profile are designated according to the standard with the letter R. In the reported article, all the parameters for evaluating the geometric structure (texture) of the machined surface were calculated on the basis of microtopography (a set of profiles representing a representation of a fragment of the surface) and designated according to the standard with the letter S (Fig. 11, Tab. 7). For this reason, it seems justified to use the term "surface texture parameters" in the submitted article.

Reviewer comment 3:

The data in Figure 5 and Figures 6-8 is the same results, it is better to delete Figure 5 or add it into supplementary material.

Authors response:

Figure 5 was removed from the article as suggested by the reviewer.

Reviewer comment 4:

The data in Table 7 and Figures 12-15 is the same results, it is better to delete Figures 12-15 or add them into supplementary material.

Authors response:

In order to avoid duplication of information, it was decided to remove Table 7 from the text of the article (instead of removing Figures 12-15). The authors assumed that charts would be a more readable form of presentation of these results for readers.

Reviewer comment 5:

Table 5 is not necessary, the abbreviation B126, B181, and B251, can be added into Table 4.

Authors response:

In the revised version of the article, Table 5 was removed and information about the abbreviated designation of grinding wheels was added to Table 4 in accordance with the reviewer's comment. In addition, Table 6 has been removed and the information from this table were placed directly on the modified version of Figure 11 (Figure 10 in revised version of manuscript).

Reviewer comment 6:

"cubic boron nitride cBN grains" should be "cubic boron nitride (cBN) grains" or "cBN grains"

Authors response:

All appearances in the text of the quoted phrase have been modified in accordance with the reviewer's comment.

Reviewer comment 7:

The first long sentence in Section 2 is not clear.

Authors response:

The goal of the experimental study has been rewritten to increase the readability of the information provided.

Reviewer 2 Report

This paper is on experimental studies of cubic BN as cutting materials. The effects of various cutting parameters on the properties of vitrified BN were examined. Lots of figures and graphs were presented to show the experimental results. The following modifications may be made.

1. The KIc should be called as "Critical stress intensity factor" or "fracture toughness" not "Critical stress intensity ratio".

2. So many tables and data figures are in the paper. Some data are presented repeatedly. It is suggested that only significant or representative data or results may be shown.

3. The first several sentences in the Abstract are not needed. They are more like introduction.

4. On line 173, "New Taipei City, Taiwan [20-22]" should be "New Taipei City, Taiwan, China [20-22]". On line 368, the country name is also missing; " New Taipei City, Taiwan" should be " New Taipei City, Taiwan, China".

5. What is exactly a vitrified BN material?

6. Are there any scanning electron microscopic images taken for the c-BN cutting material?

Author Response

The Authors wish to thank all the Reviewers for their time spent on prepare review of the manuscript “Effect of the granularity of cubic boron nitride vitrified grinding wheels on the planar technical blades sharpening process”. All the valuable comments, suggestions and hints were very helpful in improving the readability of the text and in improving of its scientific quality. Below the detailed responses upon these subsequent comments were given. The modified or added text in the manuscript was highlighted in red.

REVIEWER 2

This paper is on experimental studies of cubic BN as cutting materials. The effects of various cutting parameters on the properties of vitrified BN were examined. Lots of figures and graphs were presented to show the experimental results. The following modifications may be made.

Reviewer comment 1:

The KIc should be called as "Critical stress intensity factor" or "fracture toughness" not "Critical stress intensity ratio".

Authors response:

The indicated nomenclature error was corrected in the revised version of the article.

Reviewer comment 2:

So many tables and data figures are in the paper. Some data are presented repeatedly. It is suggested that only significant or representative data or results may be shown.

Authors response:

In the revised version of the article, instances of duplicate information have been removed (Tables 5, 6 and 7 as well as Figure 5 were deleted) in accordance with the reviewer's comment. The information from this Table 6 were placed directly on the modified version of Figure 11 (Figure 10 in revised version of manuscript)

Reviewer comment 3:

The first several sentences in the Abstract are not needed. They are more like introduction.

Authors response:

The abstract was shortened in the revised version of the article in accordance with the reviewer's comment.

Reviewer comment 4:

On line 173, "New Taipei City, Taiwan [20-22]" should be "New Taipei City, Taiwan, China [20-22]". On line 368, the country name is also missing; " New Taipei City, Taiwan" should be " New Taipei City, Taiwan, China".

Authors response:

The information on the country of origin of the microscope manufacturer was corrected in the revised version of the article.

Reviewer comment 5:

What is exactly a vitrified BN material?

Authors response:

The indicated phrase referred to the information that the grinding wheel was made of cBN abrasive grains and a vitrified bond. The text of the revised manuscript was corrected to make the information provided more unambiguous.

Reviewer comment 6:

Are there any scanning electron microscopic images taken for the c-BN cutting material?

Authors response:

The authors did not register during the research SEM microscopic images of cBN grinding wheels. However, we thank you for this question, as it makes us realize that an in-depth analysis of cBN abrasive grain wear phenomena based on SEM images is an intervening issue and should be done in further research work.

Round 2

Reviewer 1 Report

The revised manuscript has been carefully modified according to the reviewer's comments and suggestions, it could be accepted for publication on Materials.